# Vancouver Welcomes You!
# Minimalist Location Metonymy Resolution

## Abstract

Named entities are frequently used in a metonymic manner. They serve as references to related entities such as people and organisations. Accurate identification and interpretation of metonymy can be directly beneficial to various NLP applications, such as Named Entity Recognition and Geographical Parsing. Until now, metonymy resolution (MR) methods mainly relied on parsers, taggers, dictionaries, external word lists and other hand-crafted lexical resources. We show how a minimalist neural approach combined with a novel predicate window method can achieve state-of-the-art results on the SemEval 2007 task on Metonymy Resolution. Additionally, we contribute with a new Wikipedia-based MR dataset called *RelocaR*, which is tailored towards locations as well as improving previous deficiencies in annotation guidelines.

## 1 Introduction

In everyday language, we come across many types of figurative speech. These irregular expressions are understood with little difficulty by humans but require special attention in NLP. One of these is metonymy, a type of common figurative language, which stands for the substitution of the concept, phrase or word being meant with a semantically related one. For example, in "**Moscow** traded gas and aluminium with **Beijing**.", both location names were substituted in place of governments.

Named Entity Recognition (NER) taggers have no provision for handling metonymy, meaning that this frequent linguistic phenomenon goes largely undetected within current NLP. Classi-

fication decisions presently focus on the entity using features such as orthography to infer its word sense, largely ignoring the context, which provides the strongest clue about whether a word is used metonymically. A common classification approach is choosing the N words to the immediate left and right of the entity or the whole paragraph as input to the model. However, this "greedy" approach also processes input that should in practice be ignored.

Metonymy is problematic for applications such as Geographical Parsing (GP) (see a survey by Monteiro et al. (2016)) and other information extraction tasks in NLP. In order to accurately identify and ground location entities, for example, we must recognise that metonymic entities constitute false positives and should not be treated the same way as regular locations. For example, in "**London** voted for the change.", London refers to the concept of "people" and should not be classified as a location. There are many types of metonymy (Shutova et al., 2013), however, in this paper, we primarily address metonymic location mentions with reference to GP and NER.

Contributions: **1.** We investigate how to improve classification tasks by introducing a novel minimalist method called Predicate Window (PreWin), which is highly discriminating with its selection of input (achieved SOTA on SemEval 2007 MR task). PreWin outperforms other systems, which use many external features and tools. **2.** We also improve the annotation guidelines in MR and contribute with a new Wikipedia-based MR dataset called ReLocaR to address the training data shortage. **3.** We also make an annotated subset of the CoNLL 2003 (NER) Shared Task available for extra MR training data, alongside models, tools and other data for full replicability.

## 2 Related Work

Some of the earliest work on MR that used an approach similar to our method (machine learning and dependency parsing) was by Nissim and Markert (2003a). The decision list classifier with backoff was evaluated using syntactic head-modifier relations, grammatical roles and a thesaurus to overcome data sparseness and generalisation problems. However, the method was still limited for classifying unseen data. Our method uses the same paradigm but adds more features, a different machine learning architecture and a better usage of the parse tree structure.

Much of the later work on MR comes from the SemEval 2007 Shared Task 8 (Markert and Nissim, 2007) and later (Markert and Nissim, 2009). The feature set from (Nissim and Markert, 2003a) was updated to include: grammatical role of the potentially metonymic word (PMW) (such as subj, obj), lemmatised head/modifier of PMW, determiner of PMW, grammatical number of PMW (singular, plural), number of words in PMW and number of grammatical roles of PMW in current context. The winning system by Farkas et al. (2007) used these features and a maximum entropy classifier to achieve 85.2% accuracy. This was also the "leanest" system but still made use of feature engineering and some external tools. Brun et al. (2007) achieved 85.1% accuracy using local syntactical and global distributional features generated with an adapted, proprietary Xerox deep parser. This was the only unsupervised approach, based on using syntactic context similarities calculated on large corpora such as the the British National Corpus (BNC) with 100M tokens.

Nastase and Strube (2009) used a Support Vector Machine (SVM) with handcrafted features (in addition to the features provided in Markert and Nissim (2007)) including grammatical collocations extracted from the BNC to learn selectional preferences, WordNet 3.0, Wikipedia's category network, whether the entity "has-a-product" such as Suzuki and whether the entity "has-an-event" such as Vietnam (both obtained from Wikipedia). The bigger set of around 60 features and leveraging global (paragraph) context enabled them to achieve 86.1% accuracy. Once again, we draw attention to the extra training, external tools and additional feature generation.

Similar recent work by Nastase and Strube (2013), extending Nastase et al. (2012) involved transforming Wikipedia into a large-scale multilingual concept network called WikiNet. By building on Wikipedia's existing network of categories and articles, their method automatically discovers new relations and their instances on a large scale. As one of their extrinsic evaluation tasks, metonymy resolution was tested. Global context (whole paragraph) was used to interpret the target word. Using an SVM and the powerful knowledge base built from Wikipedia, the highest performance to date (a 0.1% improvement from Nastase and Strube (2009)) was achieved at 86.2%, which has remained the SOTA until now.

The related work on MR so far has made limited use of dependency trees. Typical features came in the form of a head dependency of the target entity, its dependency label and its role (subj-of-win, dobj-of-visit, etc). However, other classification tasks made good use of dependency trees. Liu et al. (2015) used the shortest dependency path and dependency sub-trees successfully to improve relation classification (new SOTA on SemEval 2010 Shared Task). Bunescu and Mooney (2005) show that using dependency trees to generate the input sequence to a model performs well in relation extraction tasks. Dong et al. (2014) used dependency parsing for Twitter sentiment classification to find the words syntactically connected to the target of interest. Joshi and Penstein-Rosé (2009) used dependency parsing to explore how features based on syntactic dependency relations can be used to improve performance on opinion mining. In unsupervised lymphoma (type of cancer) classification, (Luo et al., 2014) constructed a sentence graph from the results of a two-phase dependency parse to mine pathology reports for the relationships between medical concepts. Our methods also exploit the versatility of dependency parsing to leverage information about the sentence structure.

### 2.1 SemEval 2007 Dataset

Our main standard for performance evaluation is the SemEval 2007 Shared Task 8 (Markert and Nissim, 2007) dataset first introduced in Nissim and Markert (2003b). Two types of entities were evaluated, organisations and locations, randomly retrieved from the British National Corpus (BNC).

We only use the locations dataset, which comprises a train (925 samples) and a test (908 samples) partition. For *coarse* evaluation, the classes are **literal** (geographical territories and political entities), **metonymic** (place-for-people, place-for-product, place-for-event, capital-for-government or place-for-organisation) and **mixed** (metonymic and literal frames invoked simultaneously or unable to distinguish). The metonymic class further breaks down into two levels of subclasses allowing for *medium* and *fine* evaluation. The class distribution within SemEval is approx 80% literal, 18% metonymic and 2% mixed. This seems to be the approximate natural distribution of the classes for location metonymy, which we have also observed while sampling Wikipedia for our new dataset.

## 3 Our Approach

Our contribution broadly divides into two main parts, data and methodology. Section 3 introduces our new dataset, Section 4 introduces our new feature extraction method.

### 3.1 Design and Motivation

As part of our contribution, we created a new MR dataset called ReLocaR (Real Location Retrieval), partly due to the lack of quality annotated train/test data and partly because of the shortcomings with the SemEval 2007 dataset (see Section 3.2). Our corpus is designed to evaluate the capability of a classifier to distinguish **literal**, **metonymic** and **mixed** location mentions. In terms of dataset size, ReLocaR contains 1,000 training and 1,000 test instances. The data was sampled using Wikipedia's Random Article API[1]. We kept the sentences, which contained at least one of the places from a manually compiled list[2] of countries and capitals of the world. The natural distribution of literal versus metonymic examples is approximately 80/20 so we had to discard the excess literal examples during sampling to balance the classes.

### 3.2 ReLocaR - Improvements over SemEval

**1.** We do not break down the metonymic class further as the distinction between the subclasses is subtle and hard to agree on.

**2.** The distribution of the three classes in ReLocaR (literal, metonymic, mixed) is approximately

---

[1] https://www.mediawiki.org/wiki/API:Random
[2] Available on GitHub as locations.txt

(49%, 49%, 2%) eliminating the high bias (80%, 18%, 2%) of SemEval. We will show how such a high bias transpires in the test results (Section 5).

**3.** We have reviewed the annotation of the test partition and found that we disagreed with up to 11% of the annotations. Zhang and Gelernter (2015) disagreed with the annotation 8% of the time. Poibeau (2007) also challenged some annotation decisions. ReLocaR was annotated by 4 trained linguists (undergraduate and graduate) and 2 computational linguists (authors). Linguists were independently instructed (see section 3.3) to assign one of the two classes to each example with little guidance. We leveraged their linguistic training and expertise to make decisions rather than imposing some specific scheme. Unresolved sentences would receive the mixed class label.

**4.** The most prominent difference is a small change in the annotation scheme (after independent linguistic advice). The SemEval 2007 Task 8 annotation scheme, which can be found in (Markert and Nissim, 2007) considers the political entity interpretation a literal reading. It suggests that in "**Britain**'s current account deficit...", Britain refers to a literal location, rather than a government (which is an organisation). This is despite acknowledging that "The locative and the political sense is often distinguished in dictionaries as well as in the ACE annotation scheme...". In ReLocaR, we consider a political entity a metonymic reading.

### 3.3 Annotation Guidelines (Summary)

ReLocaR has three classes, **literal**, **metonymic** and **mixed**. Literal reading comprises territorial interpretations (the geographical territory, the land, soil and physical location) i.e. inanimate places that serve to point to a set of coordinates (where something might be located and/or happening) such as "The treaty was signed in **Italy**.", "Peter comes from **Russia**.", "**Britain**'s Andy Murray won the Grand Slam today.", "**US** companies increased exports by 50%.", "**China**'s artists are among the best in the world." or "The reach of the transmission is as far as **Brazil**.".

A metonymic reading is any location occurrence that expresses animacy (Coulson and Oakley, 2003) such as "**Jamaica**'s indifference

will not improve the negotiations.", "**Sweden**'s budget deficit may rise next year.". The following are other metonymic scenarios: a location name, which stands for any persons or organisations associated with it such as "We will give aid to **Afghanistan**.", a location as a product such as "I really enjoyed that delicious **Bordeaux**.", a location posing as a sports team "**India** beat **Pakistan** in the playoffs.", a governmental or other legal entity posing as a location "**Zambia** passed a new justice law today.", events acting as locations "**Vietnam** was a bad experience for me".

The mixed reading is assigned in two cases: either both readings are invoked at the same time such as in "The Central European country of **Slovakia** recently joined the EU." or there is not enough context to ascertain the reading i.e. both are plausible such as in "We marvelled at the art of ancient **Mexico**.". In difficult cases such as these, the mixed class is assigned.

### 3.4 Inter-Annotator Agreement

We give the IAA for the test partition only. The whole dataset was annotated by the first author as the main annotator. Two pairs of annotators (4 linguists) then labelled 25% of the dataset each for a 3-way agreement. The agreement before adjudication was 91% and 93%, 97.2% and 99.2% after adjudication (for pair one and two respectively). The other 50% of sentences were then once again labelled by the main annotator with a 97% agreement with self. The remainder of the sentences (unable to agree on among annotators even after adjudication) were labelled as a mixed class (1.8% of all sentences).

### 3.5 CoNLL 2003 and MR

We have also annotated a small subset of the CoNLL 2003 NER Shared Task data for metonymy resolution (locations only). Following the Reuters RCV1 Corpus (Lewis et al., 2004) distribution permissions[3], this data is only available by emailing the first author. There are 4,089 positive (literal) and 2,126 negative (metonymic) sentences to assist with algorithm experimentation and model prototyping. Due to the lack of annotated training data for MR, this is a valuable resource. The data was annotated by the first author, there are no IAA figures.

---

[3]http://trec.nist.gov/data/reuters/reuters.html

## 4 Methodology

### 4.1 Predicate Window (PreWin)

Through extensive experimentation and observation, we arrived at the intuition behind PreWin, our novel feature extraction method. The classification decision of the class of the target entity is mostly informed not by the whole sentence (or paragraph), rather it is a small and focused "predicate window" pointed to by the entity's head dependency. In other words, most of the sentence is not only superfluous for the task, it actually lowers the accuracy of the model due to irrelevant input. This is particularly important in metonymy resolution as the entity's surface form does not change for subsequent classifications.

In Figure 1, we show the process of extracting the Predicate Window from a sample sentence (more examples are available in the Appendix). We start by using the SpaCy dependency parser by Honnibal and Johnson (2015), which is the fastest in the world, open source and highly customisable. Each dependency tree provides the following features: dependency labels and entity head dependency. Rather than using most of the tree, we only use a single local head dependency relationship to point to the **predicate**. Leveraging a dependency parser helps PreWin with selecting the minimum relevant input to the model while discarding irrelevant input, which may cause the neural model to behave unpredictably. Finally, the entity itself is never used as input in any of our methods, we only rely on context.

PreWin then extracts up to 5 words and their dependency labels starting at the head of the entity, going in the away (from the entity) direction. The method always skips punctuation and the **conjunct** ("and", "or") relationships in order to find the predicate (see Figure 3 in the Appendix for a visual example of why this is important). The rest of the model's input is set to zeroes (see Figure 2 in the Appendix for a detailed diagram and the final model). The reason for the choice of 5 words is the balance between too much input, feeding the model with less relevant context and just enough context to capture the necessary semantics.

### 4.2 Neural Network Architecture

The output of PreWin is processed using the following machine learning model. We decided

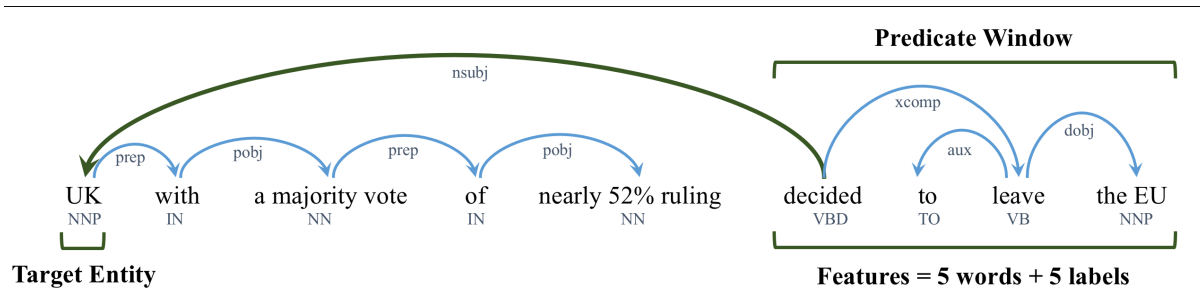

Figure 1: The predicate window starts at the head of the target entity and ends up to 4 words further, going away from the entity. The "conj" relations and punctuation are always skipped. In the above example, the head of "UK" is "decided" so PreWin takes 5 words plus labels as the input to the model. In this case, the left hand side input to the model is set to zeroes (see the Appendix for full architecture).

to use the Long Short Term Memory (LSTM) architecture by Keras[4] (Chollet, 2015). Four LSTMs are used in total, two for the left and right side (up to 5 words each) and two for the left and right dependency relation labels (up to 5 labels each). The full architecture is available in the Appendix. LSTMs are excellent at modelling language sequences (Hochreiter and Schmidhuber, 1997), (Sak et al., 2014), (Graves et al., 2013), which is why we use this type of model.

Both the Multilayer Perceptron and the Convolutional Neural Network were consistently inferior (typically 5% - 10% lower accuracy) in our earlier performance comparisons and experiments, which is also why we opted for LSTMs. For all experiments, we used a vocabulary of the first (most frequent) 100,000 word vectors in GloVe[5] Pennington et al. (2014). Finally, unless explicitly stated otherwise, the standard dimension of word embeddings was 50, which we found to work best.

### 4.3 "Immediate" Baseline

A common approach in lexical classification tasks is choosing the 5 to 10 words to the immediate right and left of the entity as input to a model such as Mikolov et al. (2013), Mesnil et al. (2013), Baroni et al. (2014) and Collobert et al. (2011). We evaluate this method (its 5 and 10-word variant) alongside PreWin and Paragraph.

### 4.4 Paragraph Baseline

The paragraph baseline method extends the "immediate" one by taking 50 words from each side of the entity as the input to the classifier. In practice,

---

[4]https://keras.io/
[5]http://nlp.stanford.edu/projects/glove/

this extends the feature window to include extrasentential evidence in the paragraph. This approach is also popular in machine learning (Melamud et al., 2016), (Zhang et al., 2016).

### 4.5 Ensemble of Models

In addition to a single best performing model, we have combined several models trained on different data and/or using different model configurations. The Ensemble method enabled us to reach SOTA results. For SemEval data, we combined two separate models (using PreWin) trained on the newly annotated 2003 CoNLL NER dataset and the training data for SemEval. For ReLocaR data, we let three models vote, all trained on the ReLocaR training data. The first model trained with 300-dimensional embeddings (PreWin method), the second used the paragraph baseline and the third used PreWin (standard 50-dimensional embeddings).

## 5 Results

We evaluate all methods using (any one or a combination of) three datasets for **training** (ReLocaR, SemEval, CoNLL) and two for **testing** (ReLocaR, SemEval).

### 5.1 Metrics and Significance

Following the SemEval 2007 convention, we use two metrics to evaluate performance, accuracy and f-score (for each class). We only evaluate at the **coarse level**, which means literal versus nonliteral (metonymic and mixed are merged into one class). In terms of statistical significance, both the SemEval dataset (1,000 samples) and our accuracy improvement (although new SOTA) are too small to be significant at the 95% confidence level.

However, the accuracy improvements of PreWin over each baseline are highly statistically significant with 99.9%+ confidence.

## 5.2 Predicate Window

Tables 1 and 2 show PreWin performing consistently better than other baselines, in many instances, significantly better and with fewer words (smaller input). Compared with the 5 and 10 window "immediate" baseline, which is the common approach in classification, PreWin is more discriminating with its input. Due to the linguistic variety and the myriad of ways the target word sense can be triggered in a sentence, it is not always the case that the 5 or 10 nearest words inform us of the target entity's meaning/type. We ought to ask what else is being expressed in the same 5 to 10-word window?

Conventional classification methods (Immediate, Paragraph) can also be seen as prioritising either feature precision or feature recall. **Paragraph** maximises the input sequence size, which maximises recall at the expense of including features that are either irrelevant or mislead the model, lowering performance. **Immediate** maximises precision by using features close to the target entity at the expense of missing important features positioned outside of its small window, once again lowering performance. PreWin can be understood as an integration of both approaches. It retains high precision by limiting the size of the feature window to 5 while maximising recall by searching anywhere in the sentence, frequently outside of a limited "immediate" window.

Perhaps we can also caution against a simple adherence to Firth (1957) *"You shall know a word by the company it keeps"*. This does not appear to be the case in our experiments as the PreWin regularly performs better than the "immediate" baseline. Further prototypical examples of the method can be viewed in the Appendix. Our intuition that most words in the sentence, indeed in the paragraph do not carry the semantic information required to classify the target entity is ultimately based on evidence. Aiming to approximate human decision making, the neural model uses only a small window (which may be far away from the entity), linked to the entity via a head dependency relationship for the final classification decision.

## 5.3 Common Errors

Most of the time (typically 85% for the two datasets), PreWin is sufficient for an accurate classification. However, it does not work well in some cases.

**Discarding important context:** Sometimes the 5 or 10 word "immediate" baseline method would actually have been preferred such as in the sentence "...REF in 2014 ranked **Essex** in the top 20 universities...". PreWin discards the right-hand side input, which is required in this case for a correct classification. Since "ranked" is the head of "Essex", the rest of the sentence gets ignored and the valuable context gets lost.

**More complex semantic patterns:** Many common mistakes were due to the lack of the model's understanding of more complex predicates such as in the following sentences: " ...of military presence of **Germany**.", "**Houston** also served as a member and treasurer of the..." or "...invitations were extended to **Yugoslavia** ...". We think this is due to a lack of training data (approx 1,000 sentences per dataset). Additional examples such as "...days after the tour had exited **Belgium**." expose some of the limitations of the neural model to recognise uncommon ways of expressing a reference to a literal place. Recall that no external resources or tools were used to supplement the training/features, the model had to learn to generalise from what it has seen during training, which was limited in our experiments.

**Parsing mistakes** were less common though still present. It is important to choose the right dependency parser for the task since different parsers will often generate slightly different parse trees. We have used SpaCy[6] for all our experiments, which is a Python-based industrial strength NLP library. Sometimes, tokenisation errors for acronyms like "U.S.A." and wrongly hyphenated words may also cause parsing errors, however, this was infrequent.

## 5.4 Flexibility of Neural Model

The top accuracy figures for ReLocaR are almost identical to SemEval. The highest single model accuracy for ReLocaR was 84.3% (85.7% with Ensemble), which was within 1% of the equivalent

---

[6]https://spacy.io/

| Method | Training (Size) | Accuracy |
|--------|-----------------|----------|
| PreWin | SemEval (925) | **66.8** |
| Immediate 5 | SemEval (925) | 59.5 |
| Immediate 10 | SemEval (925) | 59.6 |
| Paragraph | SemEval (925) | 60.4 |
| PreWin | CoNLL (6,215) | **81.3** |
| Immediate 5 | CoNLL (6,215) | 78.6 |
| Immediate 10 | CoNLL (6,215) | 80.6 |
| Paragraph | CoNLL (6,215) | 78.0 |
| PreWin | ReLocaR (1,000) | **84.3** |
| Immediate 5 | ReLocaR (1,000) | 81.9 |
| Immediate 10 | ReLocaR (1,000) | 82.6 |
| Paragraph | ReLocaR (1,000) | 82.3 |
| Ensemble | ReLocaR (1,000) | **85.7** |

Table 1: Results for the ReLocaR dataset.

methods for SemEval. Both were achieved using the same methods (PreWin or Ensemble), neural architecture and size of corpora. When the model is trained on the CoNLL (NER) data, the accuracies are 79.7% and 81.3%. This shows a good degree of flexibility in our minimalist neural network. However, when the model trained on ReLocaR and tested on SemEval (and vice versa), accuracy drops to between 66.8% and 72.5% showing that what was learnt does not seem to transfer well to another dataset. We think the reason for this is the difference in annotation guidelines; the government is a **metonymic** reading, not a literal one. This causes the model to make more mistakes.

| Method | Training (Size) | Accuracy |
|--------|-----------------|----------|
| PreWin | SemEval (925) | **85.0** |
| Immediate 5 | SemEval (925) | 81.5 |
| Immediate 10 | SemEval (925) | 81.8 |
| Paragraph | SemEval (925) | 81.3 |
| PreWin | CoNLL (6,215) | **79.7** |
| Immediate 5 | CoNLL (6,215) | 77.8 |
| Immediate 10 | CoNLL (6,215) | 78.2 |
| Paragraph | CoNLL (6,215) | 79.0 |
| PreWin | ReLocaR (1,000) | **72.5** |
| Immediate 5 | ReLocaR (1,000) | 64.8 |
| Immediate 10 | ReLocaR (1,000) | 66.0 |
| Paragraph | ReLocaR (1,000) | 66.3 |
| Nastase et al. (2013) | SemEval (1,000) | 86.2 |
| Ensemble PreWin | SemEval & CoNLL | **86.3** |

Table 2: Results for the SemEval dataset.

## 5.5 Ensemble Method

The highest accuracy and f-scores were achieved with the ensemble method for both datasets. We combined two models (previously described in section 4.5) for SemEval to achieve 86.3% accuracy (previous SOTA 86.2%) and three models for ReLocaR to achieve 85.7% for the new dataset. Training separate models with different parameters and/or on different datasets does increase classification capability as various models learn distinct aspects of the task. Letting them vote for the final label enabled the 1.3 - 1.4% improvement.

## 5.6 Dimensionality of Word Embeddings

We found that increasing dimension size (to 100 and 300) did not improve performance, in fact, accuracy decreased by an average of 3%. The neural network tended to overfit, even with fewer epochs, the results were inferior to our default 50-dimensional embeddings. We posit that fewer dimensions of the distributed word representations force the abstraction level higher as the meaning of words must be expressed more succinctly. We think this helps the model generalise better, particularly for smaller datasets. Lastly, learning word embeddings from scratch on datasets this small (1,000 samples) is possible but impractical, the performance typically decreases 5% if word embeddings are not initialised first.

| Dataset / Method | Literal | Non-Literal |
|------------------|---------|-------------|
| SemEval / PreWin | **91.9** | **60.6** |
| SemEval / SOTA | 91.6 | 59.1 |
| ReLocaR / PreWin | 86.1 | 85.6 |

Table 3: Per class f-scores - the old versus new SOTA figures (all using the Ensemble method). Note the model class bias for SemEval.

## 5.7 F-Scores and Class Imbalance

Table 3 shows the new SOTA f-scores using the Ensemble method, the previous SOTA on SemEval and the best f-scores for ReLocaR. The class imbalance inside SemEval (80% literal, 18% metonymic, 2% mixed) is reflected as a high bias in the final model. This is not the case with ReLocaR and its 49% literal, 49% metonymic and 2% mixed ratio of 3 classes. The model was equally capable of distinguishing between literal and non-literal cases.

## 5.8 Another baseline

There was another baseline we tested, however, it was not covered anywhere so far because of its low performance. It was a type of extreme parse tree pruning, during which most of the sentence gets discarded and we only retain 3 to 4 content words. The method uses non-local (long range) dependencies to construct a short input sequence. However, the method was a case of ignoring too many relevant words and accuracy was fluctuating in the mid-60% range, which is why we did not report the results. However, it serves to further justify the choice of 5 words as the predicate window as fewer words caused the model to underperform.

## 6 Discussion

### 6.1 NER, GP and Metonymy

We think the next frontier is a NER tagger, which actively handles metonymy. The task of labelling entities should be mainly driven by context rather than the word's surface form. If the target entity looks like "**London**", this should not mean the entity is automatically a location. Metonymy is a frequent linguistic phenomenon and could be handled by NER taggers to enable many innovative downstream NLP applications.

Geographical Parsing is a pertinent use case. In order to monitor/mine text documents for geographical information only, the current NER technology does not have a solution. We think it is incorrect for any NER tagger to label "Vancouver" as a location in "*Vancouver welcomes you.*". A better output might be something like the following: *Vancouver = location AND metonymy = True*. This means Vancouver is usually a location but is used metonymically in this case. How this information is used will be up to the developer. Organisations behaving as persons, share prices or products are but a few other examples of metonymy.

### 6.2 Simplicity and Minimalism

Previous work in MR such as most of the SemEval 2007 participants (Farkas et al. (2007), Nicolae et al. (2007), Leveling (2007), (Brun et al., 2007), (Poibeau, 2007)) and the more recent contributions used a selection of many of the following features/tools for classification: handmade trigger word lists, WordNet, VerbNet, FrameNet, extra features generated/learnt from parsing Wikipedia (approx 3B words) and BNC (approx 100M words), custom databases, handcrafted features, multiple (sometimes proprietary) parsers, Levin's verb classes, 3,000 extra training instances from a corpus called MAScARA [7] by Markert and Nissim (2002) and other extra resources including the SemEval Task 8 features. We managed to achieve higher performance with a small neural network, minimal training data, a basic dependency parser and the new PreWin method by being highly discriminating in choosing signal over noise.

## 7 Conclusions and Future Work

We showed how a minimalist neural approach can replace substantial external resources, handcrafted features and how the PreWin method can even ignore most of the paragraph where the entity is positioned and still achieve state of the art performance in metonymy resolution. The pressing new question is: "How much better the performance could have been if our method availed itself of the extra training data and resources used by previous works?" Indeed this is the next research chapter for PreWin.

We discussed how tasks such as Geographical Parsing can benefit from "metonymy-enhanced" NER tagging. We have also presented a case for better annotation guidelines for MR (after consulting with a number of linguists), which now means that a government is not of a literal class, rather it is a metonymic one. We fully agreed with the rest of the previous annotation guidelines. We also introduced ReLocaR, a new corpus for (location) metonymy resolution and encourage researchers to make effective use of it (including the additional CoNLL 2003 data subset we annotated for metonymy).

Further future work will be to test the PreWin method on an NER task to see if and how it can generalise to a different classification task and how the results compare to the SOTA and similar methods such as Collobert et al. (2011) using the CoNLL 2003 NER datasets. Word Sense Disambiguation (Yarowsky, 2010), (Pilehvar and Navigli, 2014) with neural networks (Yuan et al., 2016) is another related classification task suitable for testing PreWin. If it does perform better, this will be of considerable interest to classification research (and beyond) in NLP.

---

[7]http://homepages.inf.ed.ac.uk/mnissim/mascara/

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

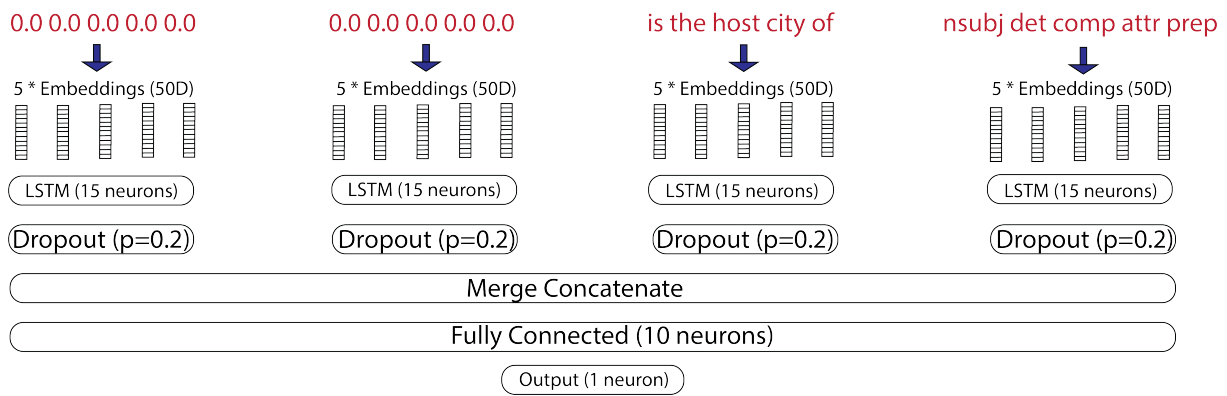

Figure 2: The neural architecture of the final model (using www.keras.io). The sentence is *Vancouver is the host city of the ACL 2017.* Small, separate sequential models are merged and trained as one. The 50-dimensional embeddings were initiated using GloVe. The *right hand* input is processed ←, the *left hand* input is processed →. This is to emphasise the importance of the words **closer** to the entity.

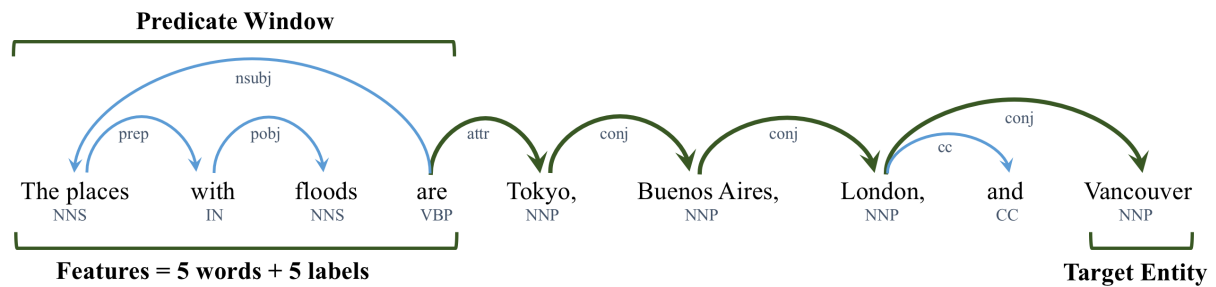

Figure 3: Why it is important for PreWin to always skip the **conjunct** dependency relation.

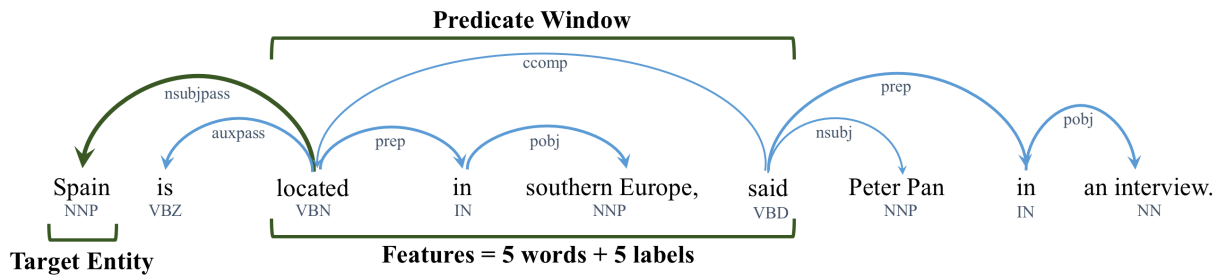

Figure 4: A lot of irrelevant input is skipped such as "is" and "Peter Pan in an interview.".

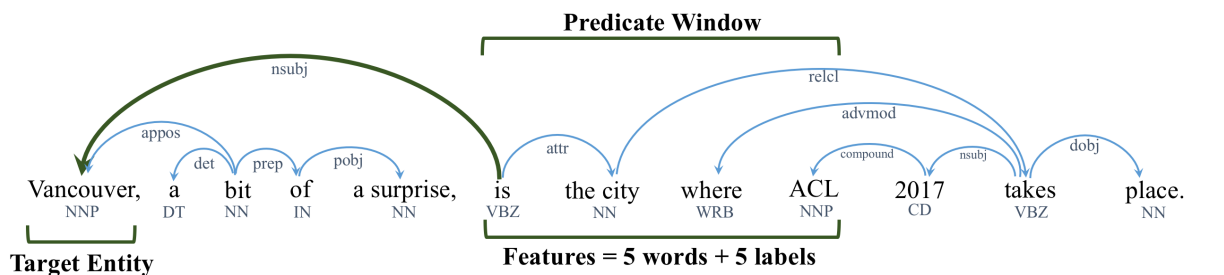

Figure 5: By looking for the predicate window, the model skips many irrelevant words.

