# Peer review of "Vancouver Welcomes You! Minimalist Location Metonymy Resolution"

_ACL 2017 — decision unknown_

[Official Review · Reviewer 1 · rating 5 · confidence 4]
soundness 3 · originality 3 · clarity 5 · impact 3 · substance 4 · appropriateness 5 · meaningful comparison 3 · presentation format Oral Presentation

- Strengths: Great paper: Very well-written, interesting results, creative
method, good and enlightening comparisons with earlier approaches. In addition,
the corpus, which is very carefully annotated, will prove to be a valuable
resource for other researchers. I appreciated the qualitative discussion in
section 5. Too many ML papers just give present a results table without much
further ado, but the discussion in this paper really provides insights for the
reader. 

- Weaknesses: In section 4.1, the sentence "The rest of the model’s input is
set to zeroes..." is quite enigmatic until you look at Figure 2. Some extra
sentence here explaining what is going on would be helpful. Furthermore, in
Figure 2, in the input layers to the LSTMs it says "5*Embeddings(50D)" also for
the networks taking dependency labels as input. Surely this is wrong? (Or if it
is correct, please explain what you mean). 

- General Discussion: Concerning the comment in 4.2 "LSTMs are excellent at
modelling language sequences ... which is why we use this type of model.". This
comment seems strange to me. This is not a sequential problem in that sense.
For each datapoint, you feed the network all 5 words in an example in one go,
and the next example has nothing to do with the preceding one. The LSTM
architecture could still be superior, of course, but not for the reason you
state. Or have I misunderstood something? I'd be interested to hear the
authors' comments on this point.